# Allosteric fine-tuning of the conformational equilibrium poises the chaperone BiP for post-translational regulation

Lukasz Wieteska[1], Saeid Shahidi[1,2†], Anastasia Zhuravleva[1]*

[1]School of Molecular and Cellular Biology, Faculty of Biological Sciences, University of Leeds, Leeds, United Kingdom; [2]Astbury Centre for Structural Molecular Biology, University of Leeds, Leeds, United Kingdom

**Abstract** BiP is the only Hsp70 chaperone in the endoplasmic reticulum (ER) and similar to other Hsp70s, its activity relies on nucleotide- and substrate-controllable docking and undocking of its nucleotide-binding domain (NBD) and substrate-binding domain (SBD). However, little is known of specific features of the BiP conformational landscape that tune BiP to its unique tasks and the ER environment. We present methyl NMR analysis of the BiP chaperone cycle that reveals surprising conformational heterogeneity of ATP-bound BiP that distinguishes BiP from its bacterial homologue DnaK. This unusual poise enables gradual post-translational regulation of the BiP chaperone cycle and its chaperone activity by subtle local perturbations at SBD allosteric 'hotspots'. In particular, BiP inactivation by AMPylation of its SBD does not disturb Hsp70 inter-domain allostery and preserves BiP structure. Instead it relies on a redistribution of the BiP conformational ensemble and stabilization the domain-docked conformation in presence of ADP and ATP.
DOI: https://doi.org/10.7554/eLife.29430.001

*For correspondence:
a.zhuravleva@leeds.ac.uk

Present address: †Princess Margaret Cancer Centre, University Health Network, Toronto, Canada

Competing interests: The authors declare that no competing interests exist.

## Introduction

The endoplasmic reticulum (ER) is an essential organelle in eukaryotic cells responsible for folding and maturation of the majority of secreted and membrane proteins. An immunoglobulin heavy-chain binding protein (BiP; also known as GRP78 and HSP5A) is the only Hsp70 chaperone in the ER. BiP binds to the majority of unfolded and misfolded proteins in this organelle to promote their folding and prevent aggregation (*Araki and Nagata, 2011*; *Vincenz-Donnelly and Hipp, 2017*).

BiP belongs to the highly conserved Hsp70 chaperone family and shares about 60% of its sequence identity with *E. coli* Hsp70 (DnaK) and human cytosolic Hsp70s (*Wang et al., 2017*). BiP and other Hsp70s have identical domain organization (*Behnke et al., 2015*) (*Figure 1*): They consist of a C-terminal substrate-binding domain (SBD) and an N-terminal nucleotide-binding domain (NDB), which are attached by a highly conserved hydrophobic linker. The SBD, which in turn, is made of a β-sandwich domain (βSBD) and a α-helical lid (αLid), binds to unfolded protein substrates, while the NBD is an ATPase that regulates the affinity of substrate binding to the SBD by using energy from ATP hydrolysis (*Mayer et al., 2001*; *Mayer, 2013*).

The majority of Hsp70 functions rely on their ability to cycle between several functional conformations (*Mayer, 2013*; *Mayer and Kityk, 2015*; *Zuiderweg et al., 2013*). Our mechanistic understanding of the Hsp70 functional cycle comes mainly from structural and dynamic characterization of bacterial Hsp70, DnaK. In DnaK, ATP binding favors a compact, domain-docked, linker-bound conformation, which has low ATPase activity (*Zhuravleva et al., 2012*; *Kityk et al., 2012*; *Qi et al., 2013*). Substrate binding to this state stabilizes a transient domain-undocked, linker-bound state,

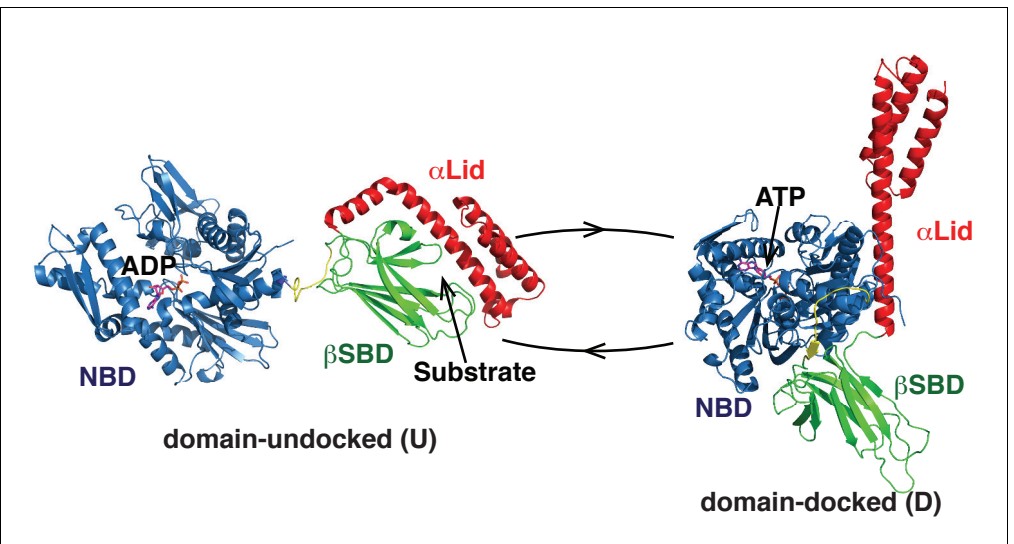

**Figure 1.** Two main functional conformations of BiP. Ribbon representation of the structure of the ATP-bound BiP (PDB 5e84 [(*Yang et al., 2015*)]), referred as the domain-docked conformation, and ADP-bound, which was modeled from the structures of the isolated NBD (PDB 5evz [*Hughes et al., 2016*]) and (D) SBD (PDB 5e85 [*Yang et al., 2015*]) and referred as the domain-undocked (U) conformation. The NBD (blue), βSBD (green), and αLid are highlighted by different colors and labeled.
DOI: https://doi.org/10.7554/eLife.29430.002

The following figure supplement is available for figure 1:

**Figure supplement 1.** Sequence alignment of eukaryotic and bacterial Hsp70 proteins.
DOI: https://doi.org/10.7554/eLife.29430.003

which has enhanced ATPase activity, efficient (fast) substrate binding and release kinetics, and low substrate affinity (*Zhuravleva et al., 2012*; *Lai et al., 2017*). ATP hydrolysis (converting ATP to ADP), results in stabilization of the domain-undocked, linker-unbound state, which has high substrate affinity, but very slow and inefficient substrate binding and release (*Bertelsen et al., 2009a*; *Swain et al., 2007*). In turn, ADP-to-ATP exchange converts the chaperone back to the transient ATP- and substrate-bound conformation that results in substrate release in the ATP-bound state so that the entire cycle can start again.

In the domain-undocked (ADP-bound) conformation, the two Hsp70 domains behave independently and are structurally identical to the isolated NBD and SBD (*Zhuravleva et al., 2012*; *Bertelsen et al., 2009a*; *Swain et al., 2007*) (*Figure 1*, left). Remarkably, for a number of different Hsp70s, X-ray and NMR structures of the isolated domains (including those for BiP) are almost identical, independent of ligand-bound status and organism (*Mayer and Kityk, 2015*; *Wisniewska et al., 2010*; *Zhang et al., 2014*). Moreover, recent X-ray structures of ATP-bound DnaK (*Kityk et al., 2012*; *Qi et al., 2013*) and human BiP (*Yang et al., 2015*) revealed only subtle differences in the domain-docked conformation for these two chaperones: In both structures the interdomain linker binds to the NBD; the βSBD and NBD are docked to each other with the αLid detached from the βSBD and interacting with the NBD; whilst the βSBD is opened, making the βSBD pocket solvent accessible and ready to bind to a client protein (*Yang et al., 2015*) (*Figure 1*, right). Despite this structural similarity, the functional activity of different Hsp70s, including DnaK and BiP, varies very significantly (*Behnke et al., 2015*; *Bonomo et al., 2010*) in response to differences in the protein-folding environment for different Hsp70s and the wide range of variations in substrates and their specificity. Consequently, the fundamental molecular mechanisms that underlie these functional differences without significant structural variations between different Hsp70s remain unknown.

Growing evidence has suggested that Hsp70 chaperones are extremely flexible in solution and crystallographic data have provided only 'end-point' snapshots of the chaperone landscape (*Zhuravleva et al., 2012*; *Lai et al., 2017*; *Zhuravleva and Gierasch, 2015*; *Zhuravleva and Gierasch, 2011*; *Marcinowski et al., 2011a*; *Bhattacharya et al., 2009*). In turn, transitions between

these 'end-point' states and chaperone conformational dynamics have been recently hypothesized to be key factors that control the Hsp70 functional cycle (*Zhuravleva et al., 2012*; *Kityk et al., 2012*; *Zhuravleva and Gierasch, 2015*; *Kityk et al., 2015*). Indeed, a thermodynamic distribution between different DnaK conformations, and thus its ATPase activity and substrate binding and release, are under precise adjustments by regulatory regions (allosteric 'hotspots') that enable regulation of DnaK conformational transitions. Even subtle perturbations (such as amino-acid substitutions or ligand binding) at these allosteric hotspots have been shown to affect and fine-tune the DnaK conformational cycle and adjust its chaperone activity (*Zhuravleva et al., 2012*; *Zhuravleva and Gierasch, 2015*). Remarkably, many allosteric hotspots are not fully conserved (*Figure 1—figure supplement 1*), suggesting a plausible molecular mechanism for functional diversity within the Hsp70 family as well as for real-time regulations of Hsp70 activity post-translationally, for example, through covalent modifications and/or interactions with co-chaperones (*Zhuravleva et al., 2012*; *Kityk et al., 2012*; *Zhuravleva and Gierasch, 2015*; *Kityk et al., 2015*; *Ahmad et al., 2011*; *Mapa et al., 2010*).

However, until now our ability to assess this hypothesis has been hampered by limited experimental information on the chaperone cycle of eukaryotic Hsp70s (*Marcinowski et al., 2011a*; *Mapa et al., 2010*; *Sikor et al., 2013*). In this study, we addressed this challenge and deployed methyl nuclear magnetic resonance (NMR) and isothermal titration calorimetry (ITC) to characterize the allosteric cycle for the ER member of the Hsp70 family BiP. We identified key conformational features that distinguish this ER chaperone from DnaK (and potentially other Hsp70s), and elucidated how the BiP conformational landscape is regulated posttranslationally by AMPylation of its βSBD.

## Results

### Construct design to 'trap' individual steps of the BiP allosteric cycle

To characterize individual steps of the BiP chaperone cycle, we used ITC to obtain thermodynamic fingerprints of nucleotide binding and induced conformational transitions, as well as methyl NMR to monitor conformational changes with atomistic details (*Tugarinov and Kay, 2005*; *Ruschak and Kay, 2010*; *Rosenzweig and Kay, 2014*). All of the experiments were performed using a well-established variant of wild-type (WT) BiP (referred as BiP* in this paper), in which Thr 229 was mutated to Gly to slow down protein ATPase activity (*Wei et al., 1995*). This substitution avoids complications for the ATP-bound BiP coming from sample heterogeneity upon ATP hydrolysis and allowed us to 'trap' BiP in the ATP-bound conformation. The T229G variant of BiP has been extensively characterized in the literature and retains WT-like conformational features (*Wei et al., 1995*; *Wei and Hendershot, 1995*). To monitor BiP conformational changes in the presence of different ligands and distinguish long-range conformational perturbations from local effects of ligand binding, we used the 'divide-and-conquer' approach (*Ruschak and Kay, 2010*; *Gelis et al., 2007*) that we had previously adopted for characterization of the *E. coli* BiP homolog DnaK (*Zhuravleva et al., 2012*). To 'trap' and characterize individual conformational transitions, we compared full-length (FL), two-domain BiP with its isolated nucleotide-binding domain (T229G variant, referred as NBD*). To elucidate the role of the linker in BiP conformational transitions, we compared the NBD* constructs with and without the interdomain linker (residues 1–417 and residues 1–413, respectively).

### Thermodynamic features of the BiP allosteric cycle

Our ITC data (*Table 1*) showed that nucleotide binding to BiP occurred with weaker µM affinity than for DnaK and in general agreed with previous observations (*Wei and Hendershot, 1995*; *Macias et al., 2011*; *Lamb et al., 2006*). In our hands, ATP binding to BiP* was near 7-fold more preferable than ADP binding (*Table 1*). Interestingly, upon truncation of the interdomain linker and SBD, ATP affinity of the isolated NBD* decreased more than nine times as compared with FL BiP*. However, the truncation of the SBD only (i.e., without the linker truncation) results in a diminished effect (*Table 1*). Taken together, these observations suggested an important stabilizing role of domain communication and/or linker-NBD interactions for ATP binding.

ATP (but not ADP) binding to FL BiP* was entropically driven with the contribution from the entropy (-TΔS) to the binding energy of about −19–20 kcal/mol (*Table 1*). This favorable entropic contribution to the ATP binding energy has been previously observed for an ATPase deficient variant

**Table 1.** Thermodynamics of nucleotide binding for BiP and DnaK.
ITC experiments for full-length (FL) BiP* and its two NBD constructs with and without the interdomain linker, NBD(1-413) and NBD(1-417), were repeated three times and the standard deviations for all parameters were less than 10%.

| | ANP | $K_d$ (µM) | $\Delta G$ (kcal/mol) | $\Delta H$ (kcal/mol) | $-T\Delta S$ (kcal/mol) |
|---|---|---|---|---|---|
| DnaK* FL | ATP | 0.16 | −9.3 | 9.0 | −18.3 |
| | ADP | 0.25 | −9.0 | −3.2 | −5.8 |
| DnaK* NBD | ATP | 0.61 | −8.9 | −4.1 | −3.7 |
| | ADP | 0.17 | −9.2 | −3.3 | −6.0 |
| BiP* FL | ATP | 0.80 | −8.3 | 11.4 | −19.7 |
| | ADP | 5.73 | −7.1 | −7.4 | 0.3 |
| BiP* NBD(1-413) | ATP | 7.41 | −7.0 | 12.3 | −19.3 |
| | ADP | 5.27 | −7.2 | −7.4 | 0.2 |
| BiP* NBD(1-417) | ATP | 1.15 | −8.1 | 14.7 | −22.3 |
| | ADP | 3.94 | −7.4 | −6.4 | 1.0 |

*Thermodynamic parameters for ATPase deficient DnaK (DnaK*) and its NBD construct without the interdomain linker were taken from Taneva et al (*Taneva et al., 2010*).
DOI: https://doi.org/10.7554/eLife.29430.004

of DnaK (DnaKT199A) (*Taneva et al., 2010*) and has been predominantly associated with enhanced interdomain and intradomain SBD flexibility due to αLid dissociation from βSBD and destabilization of the βSBD structure (*Zhuravleva et al., 2012*; *Zhuravleva and Gierasch, 2015*). In turn, the entropic contribution into ATP binding energy was significantly smaller for the isolated DnaK NBD, while ATP binding had a similar enthalpy driven thermodynamic pattern as ADP binding (*Taneva et al., 2010*). On the contrary, ATP binding to the isolated BiP NBD* is predominantly entropically driven (*Table 1*), suggesting that ATP binding to BiP and DnaK has different effects on chaperone dynamics. In contrast to DnaK, ATP binding to BiP resulted in enhanced linker-induced intradomain flexibility in its NBD, while ATP binding to DnaK affected the entire protein.

## Conformational features of the BiP allosteric cycle

To obtain site-specific information about large changes in protein structure such as domain docking/undocking, we used methyl NMR (*Rosenzweig and Kay, 2014*; *Manley and Loria, 2012*) and introduced NMR-active methyl groups to all Ile-δ1 positions (*Tugarinov et al., 2006*). Consistent with DnaK, (*Zhuravleva et al., 2012*; *Swain et al., 2007*; *Bertelsen et al., 2009b*) in the absence of ATP (i.e., in the ADP-bound state), we observed almost no changes in methyl NMR peak positions between FL BiP* and its isolated NBD and SBD (*Figure 2*, left). This suggests that ADP-bound BiP lacks domain communication, the two BiP domains behave independently, and the protein adopts the domain-undocked conformation (*Figure 1*, left).

ATP binding to FL BiP* drastically affected the methyl peak pattern and the resulting NMR spectrum possessed two sets of peaks with near identical peak intensities (*Figure 2*, right), suggesting that two BiP conformations, interconverting on the slow NMR time-scale, co-existed in solution. When the spectrum of ATP-bound FL BiP* was overlaid with the spectrum of the ATP-bound isolated NBD* (residues 1–413) and isolated SBD, peaks from the isolated NBD and SBD spectra were almost perfectly overlapping with one set of FL peaks. This set of peaks in the FL BiP spectrum was assigned to the ATP-bound domain-undocked (U) conformation. The second set of FL peaks had distinct chemical shifts from those observed in the spectra of the isolated NBD and SBD. This set of peaks was assigned to the ATP-bound domain-docked (D) conformation. To obtain populations of the domain-docked and -undocked BiP conformations, three representative non-overlapping NBD peak doublets were selected (*Figure 2*, grey box) and then the ratio of U and D peak intensities was calculated for each doublet (Materials and methods for details). This analysis revealed that in the presence of ATP, only ~53 ± 7% of BiP molecules adopt the domain-docked conformation, while the rest of the protein is domain-undocked.

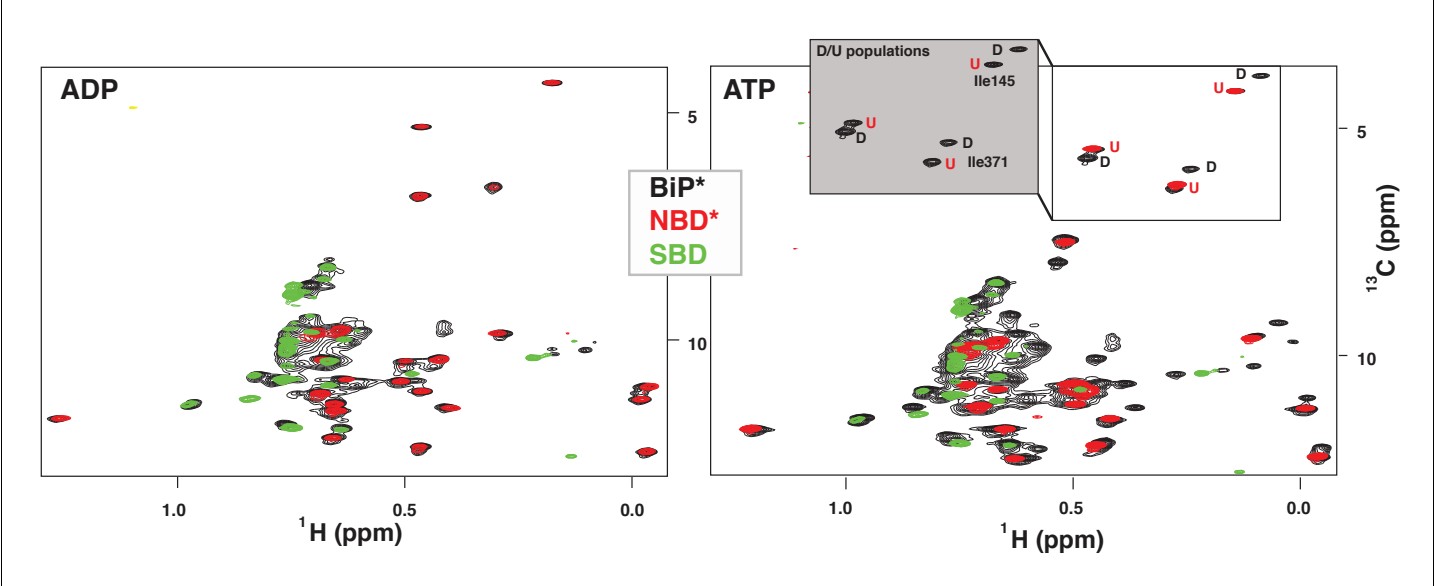

**Figure 2.** NMR fingerprints of two main functional conformations of BiP.  The isoleucine region of methyl-TROSY spectra of ADP-bound (left) and ATP-bound (right) BiP* (the full-length ATPase deficient T229G BiP construct, in black) overlaid with the spectra of corresponding nucleotide-bound state of isolated NBD* (the ATPase deficient NBD construct without the interdomain linker, residues 1–413, in red) and isolated SBD (in green). (Grey box, right) Blowup of the representative region of methyl-TROSY spectra of ATP-bound BiP*, showing three non-overlapping peak doublets, used to calculate the populations domain-docked (D) and -undocked (U) conformations. The U/D assignments details can be found in Materials and methods; briefly, FL BiP* peaks (black) that overlapping with peaks from the NBD*(1-413) spectrum (blue), were assigned to the domain-undocked conformation (labeled 'U'; that is, for the corresponding conformation the isoleucine chemical environment is very similar in the FL protein and isolated NBD); the second peak from each doublet was assigned to the domain-docked conformation (labeled 'D', that is, for the corresponding conformation the isoleucine chemical environment in the FL protein is significantly different from the chemical environment in the isolated NBD).

DOI: https://doi.org/10.7554/eLife.29430.005

The following figure supplement is available for figure 2:

**Figure supplement 1.** NMR fingerprints of the main steps of the BiP allosteric cycle.
DOI: https://doi.org/10.7554/eLife.29430.006

Taking into account that chemical shift differences between the domain-undocked and -docked conformations were less than 0.03 ppm (0.4 ppm) for $^1$H ($^{13}$C) chemical shifts and no significant line broadening was observed in the methyl NMR spectrum, we estimated that interconversion between the domain-undocked and -docked conformation is significantly slower that ~50 milliseconds (estimated as the slow exchange limit with the respect to the chemical shift differences between the two conformations). Such slow motions (and thus a large activation energy of the domain docking process) is not unexpected, given that domain docking is a multistep process that requires very large structural rearrangements (e.g., the αLid displacements are on a scale of ~10 nanometers, *Figure 1*) (*Kityk et al., 2012*; *Qi et al., 2013*). Interestingly, domain docking in DnaK is a significantly faster process as reported by a characteristic linear peak-walking pattern in DnaK NMR spectra, observed upon the gradual population of the domain-docked conformation (*Zhuravleva et al., 2012*).

Similar to DnaK, (*Zhuravleva et al., 2012*) binding of the model peptide substrate (P2: HTFPAVL [*Marcinowski et al., 2011b*]) to ATP-bound BiP* destabilized domain docking (*Figure 2—figure supplement 1*). As demonstrated previously, ATP- and substrate-bound DnaK* (a T199A ATPase deficient variant of DnaK) co-exists as a conformational ensemble of the ADP-like domain-undocked conformation with the interdomain linker exposed to the solvent and a transient linker-bound domain-undocked conformation that enables substrate interaction in the FL chaperone (*Zhuravleva et al., 2012*; *Lai et al., 2017*). In turn, ATP-induced binding of the inter-domain linker to the DnaK NBD (*Figure 3A*) has been shown to be essential for efficient ATPase activity (*Swain et al., 2007*; *Vogel et al., 2006*) and domain docking (*Zhuravleva et al., 2012*; *Lai et al., 2017*). To examine whether ATP also favors linker binding to the BiP NBD, we performed the NMR

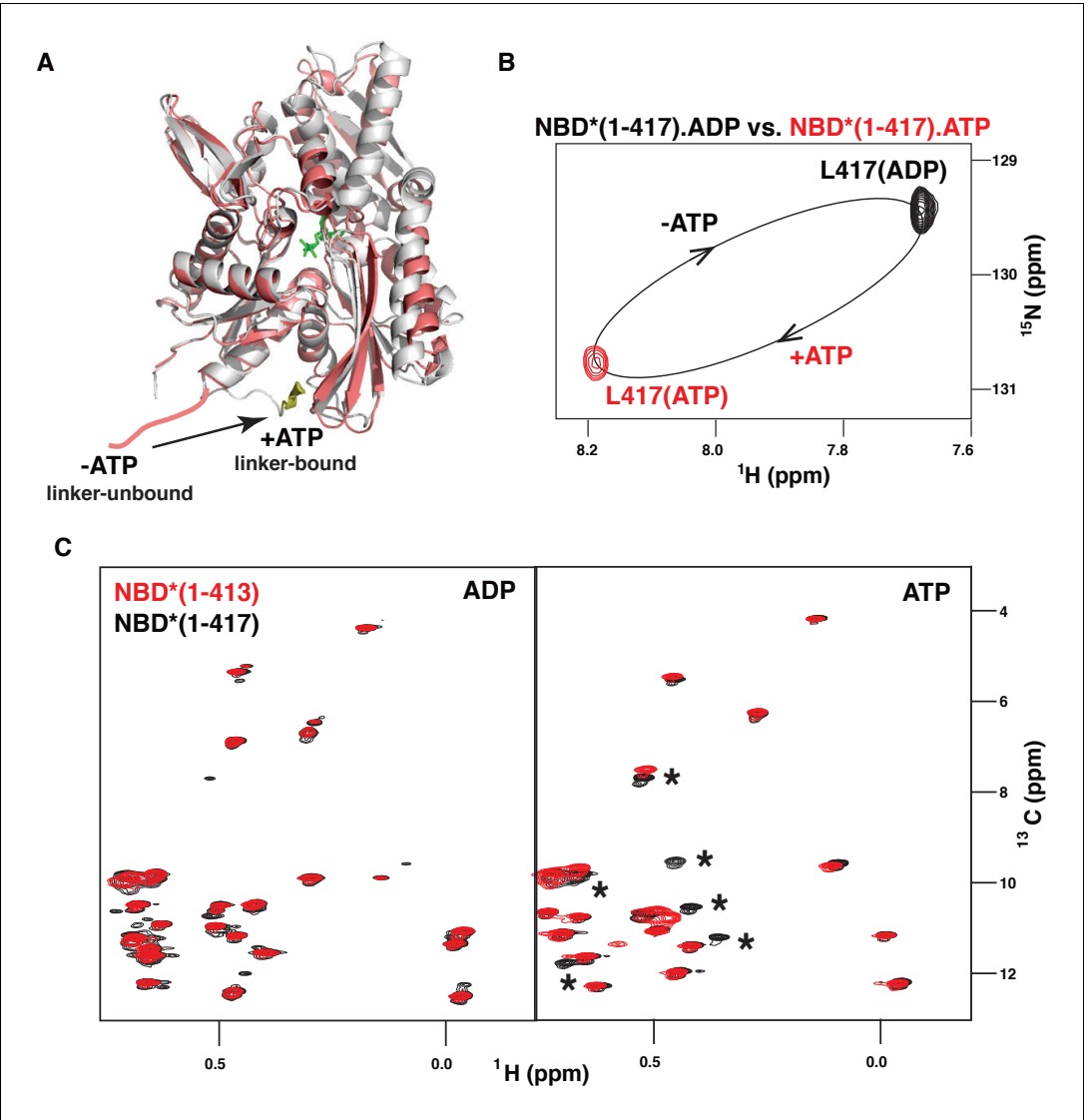

**Figure 3.** ATP-induced linker binding to the BiP NBD. (**A**) Ribbon representation of the structures of two BiP NBD conformations: linker-bound and linker-unbound. The ATP-bound NBD structure (grey, linker bound; the linker is shown in yellow) is taken from the domain-docked structure of FL BiP (PDB 5e84 [*Yang et al., 2015*]); the ADP-bound structure (pink) is taken from the ADP-bound isolated BiP NBD (residues 1–407) PDB 5evz (*Hughes et al., 2016*); for ADP-bound conformation the linker region (after residue 407) is schematically shown in pink. (**B**) Blowup of the representative region of amide TROSY of the ADP-bound (black) and ATP-bound (red) NBD*(1-417) showing the peak corresponding the C-terminal linker residue Leu417. (**C**) The isoleucine region of the methyl-TROSY spectra of the ADP- and ATP-bound BiP NBD*(1-417) (in black) overlaid with the spectra of the corresponding nucleotide-bound state of BiP NBD*(1-413) (in red). Asterisks highlight changes observed in the presence of the linker.

DOI: https://doi.org/10.7554/eLife.29430.007

analysis of two isolated NBD* constructs, NBD*(1-417) that retains the four highly conserved hydrophobic linker residues LVLL, and NBD*(1-413) that does not. Our analysis revealed that ATP-binding to the isolated BiP NBD* affects interdomain linker conformation. Indeed, ATP binding to NBD*(1-417) results in large chemical shift perturbations for the C-terminal linker residue Leu417 (*Figure 3B*). In turn, the comparison of methyl NMR spectra of NBD*(1-417) and NBD*(1-413) in the presence of ADP and ATP revealed that the LVLL linker truncation results in significant chemical shift perturbations for ~25% of isoleucine NBD* residues in the presence of ATP, but not in the presence

of ADP (*Figure 3C*). These results suggested that ATP-induced linker binding alters NBD conformation. Finally, all domain-undocked peaks in the spectrum of ATP-bound FL BiP* were perfectly overlapped with the NBD*(1-413) spectrum (*Figure 2*, right), while chemical shift perturbations were found when the spectrum of the FL protein was overlaid with the NBD*(1-417) spectrum. Taken together, these observations clearly demonstrate that similar to DnaK, ATP binding to the isolated BiP NBD favors linker interactions with the NBD and the presence of the linker and ATP results in allosteric changes in the NBD. The same two-way communication between the interdomain linker and the nucleotide-binding site was observed for the DnaK NBD, (*Zhuravleva and Gierasch, 2011*) and can be attributed to ATP-induced binding of the interdomain linker to the NBD (*Figure 3A*). However, for BiP no significant amount of the linker-bound, domain-undocked conformation was observed for the FL BiP*.

Taking together, our NMR analysis suggests that despite structural similarity to DnaK, BiP has unique thermodynamic and kinetic features of its chaperone conformational landscape, apparently implied by a cumulative effect of amino acid sequence variations between these two Hsp70s (*Figure 1—figure supplement 1*).

## Allosteric regulation of the BiP conformational ensemble

Next, we examined whether the BiP conformational equilibrium can be controlled allosterically. We introduced several 'soft' amino acid substitutions along the Hsp70 inter-domain allosteric network (*Zhuravleva et al., 2012*) connecting the substrate-binding site with the NBD-SBD interdomain interface (*Figure 4A*). We mutated Val461, located near the substrate-binding site, to Phe; Ile526, located at the central hub for the βSBD allosteric network – the β5/β7/β8 hydrophobic core, to Val; Ile437, located near the βSBD-NBD interface, to Val; and Ile538, located at the βSBD-αLid and αLid-NBD interfaces, to Val. Methyl NMR revealed that in the presence of ATP all of these 'soft' mutations affected the equilibrium between the domain-docked and -undocked conformations, as reported by redistribution of intensities for peaks corresponding to these conformations (*Figure 4B*, *Figure 4—source data 1*). Remarkably, the 'soft' mutations enables both gradual stabilization and destabilization of the domain-docked conformation (*Figure 4B*). Indeed, the V461F (*Figure 4—figure supplements 1* and *2*) and I526V (*Figure 4—figure supplement 4*) substitutions favors domain docking, while the I437V (*Figure 4—figure supplement 3*) and I538V (*Figure 4—figure supplement 5*) substitutions stabilizes the domain-undocked conformation. Moreover, even in the absence of ATP, the domain-docked conformation is significantly populated for BiP*-V461F (~49%) and BiP*-I526V (~35%) (*Figure 4C*, *Figure 4—figure supplement 4*, and *Figure 4—source data 1*), suggesting that ATP binding is not an essential step for domain docking.

## Regulation of the BiP conformational ensemble by AMPylation

Would allosteric 'tunability' of the BiP conformation equilibrium provide a plausible explanation of how BiP is post-translationally regulated in cells? To test this hypothesis, we examined how the post-translational modification of BiP by AMPylation onto Thr518 (*Preissler et al., 2015*) affects the BiP conformational cycle.

The AMPylation site Thr518 is located on the loop $L_{7,8}$ of the βSBD that connects allosterically important strands β7 and β8 (*Figure 4A*), an allosteric region that controls βSBD conformation in DnaK (*Zhuravleva and Gierasch, 2015*). In particular, perturbations in the loop $L_{7,8}$ and strands β7 and β8 regulate opening of the substrate-binding site and flexibility of the substrate-binding loops. In line with the previous observations for DnaK, the I526V substitution in strand β8 also affects the BiP conformational ensemble and partially stabilizes the domain-docked conformation even in the absence of ATP (*Figure 4B* and *Figure 4—source data 1*). Next, to examine whether AMPylation of Thr518 and the I526V substitution control BiP through the same allosteric mechanism, we calculated the populations of the domain-docked and -undocked conformations for AMPylated and non-AMPylated BiP* bound to ATP and ADP (*Figure 5A*). Our results suggested that AMPylation of the loop $L_{7,8}$ has a similar effect as the I526V substitution in the strand β8 and the V461F substitution in the substrate-binding site: It shifts the BiP conformational equilibrium toward the domain-docked conformation in the presence of ATP (68% of the docked conformation), but also significantly stabilizes the domain docking in the absence of ATP (64% of the docked conformation, *Figure 5—figure supplement 1* and *Figure 5—source data 1*).

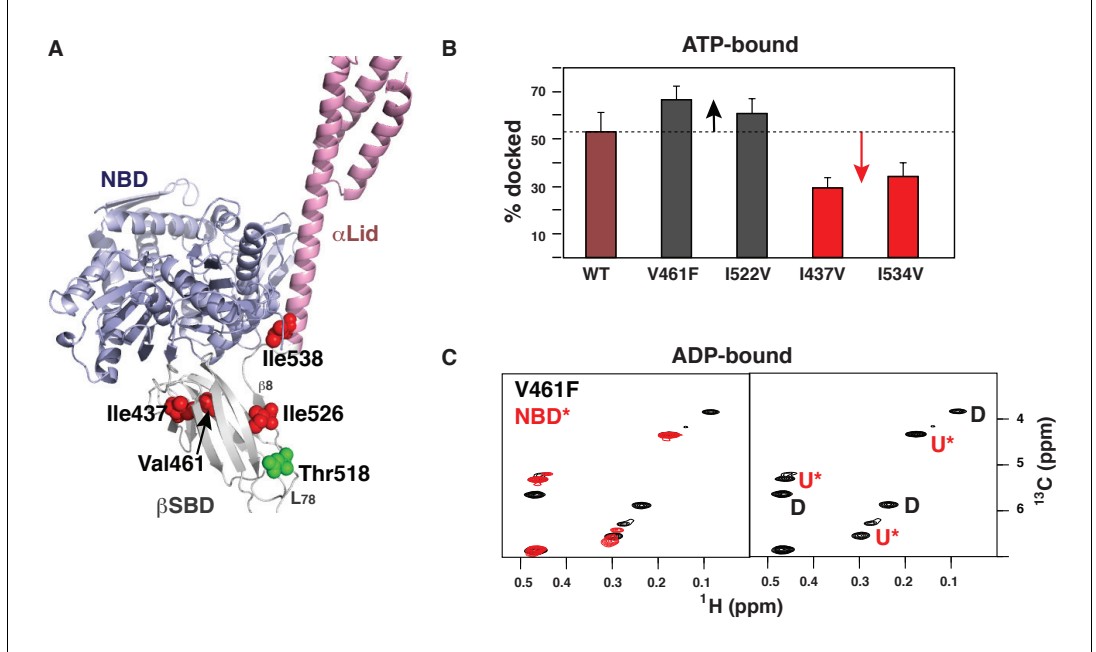

**Figure 4.** Allosteric regulation of the BiP conformational ensemble. (A) Ribbon representation of the domain-docked conformations (PDB 5e84 [*Yang et al., 2015*]). Four 'soft' mutations are shown as red spheres. The AMPylation site (Thr 518) is shown in green. (B) Populations of the domain-docked conformation calculated from methyl-TROSY spectra of ATP-bound BiP* 'soft' mutants, V461F, I526V, I437V and I538V, using methyl peak intensities of the domain-docked and -undocked conformations. Error bars show SDs from the means for three peak doublets (Materials and methods, *Figure 4—figure supplements 1–5* and *Figure 4—source data 1*). (C) Blowup of the representative region of the methyl-TROSY spectrum of ADP-bound BiP*-V461F (black) overlaid (left) with the spectrum of ADP-bound NBD*(1-413) (red). (Right) For each FL BiP* doublet, the peak overlapping with the ADP-bound NBD* peak is labeled as U* (ADP-bound, domain-undocked) and the second peak is labeled as D (ADP-bound, domain-docked).
DOI: https://doi.org/10.7554/eLife.29430.008

The following source data and figure supplements are available for figure 4:

**Source data 1.** NMR analysis of populations for the domain-docked and -undocked conformations.
DOI: https://doi.org/10.7554/eLife.29430.014

**Figure supplement 1.** The isoleucine regions of methyl-TROSY spectra of ATP- and ADP-bound full-length (FL) BiP* (A, in black), overlaid with the spectra of a corresponding nucleotide-bound state of NBD*(1-413) (B, in red).
DOI: https://doi.org/10.7554/eLife.29430.009

**Figure supplement 2.** The isoleucine regions of methyl-TROSY spectra of ATP- and ADP-bound states of the V461F variant of full-length (FL) BiP* (A, in black), overlaid with the spectra of a corresponding nucleotide-bound state of NBD*(1-413) (B, in red).
DOI: https://doi.org/10.7554/eLife.29430.010

**Figure supplement 3.** The isoleucine regions of methyl-TROSY spectra of ATP- and ADP-bound states of the I437V variant of full-length (FL) BiP* (A, in black), overlaid with the spectra of a corresponding nucleotide-bound state of NBD*(1-413) (B, in red).
DOI: https://doi.org/10.7554/eLife.29430.011

**Figure supplement 4.** The isoleucine regions of methyl-TROSY spectra of ATP- and ADP-bound states of the I526V variant of full-length (FL) BiP* (A, in black), overlaid with the spectra of a corresponding nucleotide-bound state of NBD*(1-413) (B, in red).
DOI: https://doi.org/10.7554/eLife.29430.012

**Figure supplement 5.** The isoleucine regions of methyl-TROSY spectra of ATP- and ADP-bound states of the I538V variant of full-length (FL) BiP* (A, in black), overlaid with the spectra of a corresponding nucleotide-bound state of NBD*(1-413) (B, in red).
DOI: https://doi.org/10.7554/eLife.29430.013

It has been recently demonstrated that AMPylation efficiency depended on BiP conformation (*Preissler et al., 2015*). Indeed, in our hands, AMPylation of WT BiP* was incomplete (~60%, *Figure 5—figure supplement 7*). To achieve sample homogeneity, we also performed the experiments on the V461F BiP* variant, which made AMPylation near 100% efficient (*Figure 5—figure supplement 7*). While, as discussed above, the V461F substitution itself considerably favors domain docking in the presence (~67%) and in the absence (~49%) of ATP, its AMPylation further stabilizes the domain-docked conformation (*Figure 5A*, *Figure 5—figure supplement 2* and *Figure 5—source data 1*). As a result, AMPylated BiP*-V461F predominantly (>80%) adopts the domain-docked

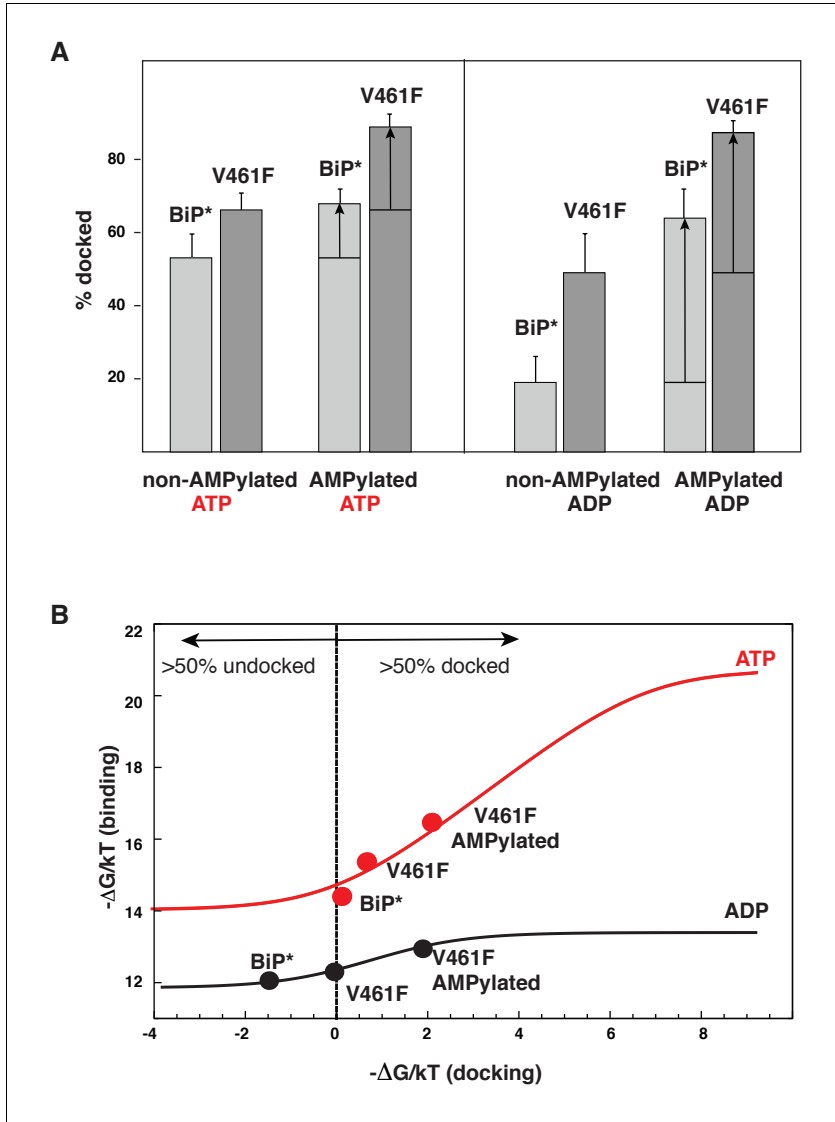

**Figure 5.** Allosteric regulation of the BiP by AMPylation. (**A**) Populations of the domain-docked conformation for AMPylated and non-AMPylated BiP* and its V461F variant in the presence of ATP and ADP, calculated using methyl peak intensities of the domain-docked and -undocked conformations. Error bars show SDs from the means for three peak doublets (Materials and methods, *Figure 5—figure supplements 1–2* and *Figure 5—source data 1*). (**B**) The thermodynamic linkage between domain docking and nucleotide binding: Plot of the experimental free energy of ATP (red dots) and ADP (black dots) binding against the experimental free energy of domain docking for BiP* and its non-AMPylated and AMPylated V461F variant. For each construct, the free energy of nucleotide binding was obtained from the ITC data and the free energy of domain docking was calculated from NMR peak intensities (see *Figure 5—source data 2* and Materials and methods for details). The red (for ATP) and black (for ADP) curves show the theoretical plots of the free energy of ATP (red) and ADP (black) binding against the free energy of domain docking under assumption that BiP co-exists as an ensemble of the domain-undocked and -docked conformations (see Materials and methods).

DOI: https://doi.org/10.7554/eLife.29430.015

The following source data and figure supplements are available for figure 5:

**Source data 1.** NMR analysis of populations for the domain-docked and -undocked conformations for AMPylated BiP.
DOI: https://doi.org/10.7554/eLife.29430.023
**Source data 2.** Analysis of the thermodynamic linkage between domain docking and nucleotide binding.
DOI: https://doi.org/10.7554/eLife.29430.024
*Figure 5 continued on next page*

*Figure 5 continued*

**Figure supplement 1.** The isoleucine regions of methyl-TROSY spectra of ATP- and ADP-bound states of AMPylated full-length (FL) BiP* (A, in black), overlaid with the spectra of a corresponding nucleotide-bound state of NBD*(1-413) (B, in red).

DOI: https://doi.org/10.7554/eLife.29430.016

**Figure supplement 2.** The isoleucine regions of methyl-TROSY spectra of ATP- and ADP-bound states of AMPylated full-length (FL) V461F BiP* (A, in black), overlaid with the spectra of a corresponding nucleotide-bound state of NBD*(1-413) (B, in red).

DOI: https://doi.org/10.7554/eLife.29430.017

**Figure supplement 3.** The methyl TROSY spectrum of ATP-bound AMPylated BiP*-V461F (black) overlaid with the spectrum of ADP-bound AMPylated BiP*-V461F (red).

DOI: https://doi.org/10.7554/eLife.29430.018

**Figure supplement 4.** The amide TROSY spectrum of ATP-bound AMPylated BiP*-V461F (black) overlaid with the spectrum of ADP-bound AMPylated BiP*-V461F (red).

DOI: https://doi.org/10.7554/eLife.29430.019

**Figure supplement 5.** Representative panels showing nucleotide (ATP or ADP) binding to BiP* and isolated NBD* (1-413) and NBD*(1-417) measured by ITC.

DOI: https://doi.org/10.7554/eLife.29430.020

**Figure supplement 6.** Representative panels showing nucleotide (ATP or ADP) binding to non-AMPylated and AMPylated V461F BiP* measured by ITC.

DOI: https://doi.org/10.7554/eLife.29430.021

**Figure supplement 7.** LC-MS data for molecular mass determination of unmodified (panels A-B) and AMPylated (panels C-D) BiP.

DOI: https://doi.org/10.7554/eLife.29430.022

conformation in the presence of either ATP or ADP (*Figure 5A*). Remarkably, no chemical shift differences were observed in methyl NMR spectra for the peaks corresponding to the BiP domain-docked conformation, regardless either the BiP* or its V461F variant were ATP-bound and non-AMPylated, ATP-bound and AMPylated or ADP-bound and AMPylated. Moreover, then the domain-docked conformation became predominantly populated for the AMPylated V461F variant, the methyl spectra of ADP-bound and ATP-bound conformation were almost identical (*Figure 5—figure supplement 3*), while only a few minor (local) chemical shift perturbations were observed in amide NMR spectra (*Figure 5—figure supplement 4*). These results are in excellent agreement with the biochemical and structural study of AMPylated BiP, where linker-specific proteolysis and X-ray crystallography was used to demonstrate that, in the absence of ATP, AMPylation stabilizes the structurally same domain-docked BiP conformation that is normally observed upon ATP binding (*Preissler et al., 2017*).

## Post-translational fine-tuning of the BiP conformational equilibrium provides the precise thermodynamic control of nucleotide binding

To examine how perturbations of the domain-docked/undocked equilibrium affects thermodynamics of nucleotide binding, we plotted the free energy of ATP and ADP binding versus the free energy difference between the domain-docked and -undocked conformations (the free energy of docking) for the BiP* (*Table 1*, *Figure 5—figure supplement 5*) and its AMPylated and non-AMPylated V461F variant (*Figure 5—figure supplement 6*, *Figure 5—source data 2*). Our results demonstrate that stabilization of the domain-docked conformation gradually increases the ADP and ATP binding energy (*Figure 5B*). This correlation between the free energies of domain docking and nucleotide binding can be quantitatively rationalized by considering kinetics and thermodynamics of the multi-step ATP-induced conformational transitions in the BiP molecule. The simplest model, in which the BiP co-exists as an ensemble of two thermodynamically distinct states, was suitable to fit the experimental data (*Figure 5B*). In this model the observed binding constant is equal to a population-weighted sum of the nucleotide binding constants for the domain-docked and -undocked conformations. The best agreement with the experimental data was achieved when ATP (ADP) binding constants were 800 nM (7 μM) and 1 nM (1.5 μM) for the domain-undocked and –docked states, respectively (*Figure 5B*).

Intriguingly, while the ADP-bound and ATP-bound domain-docked conformations are structurally almost identical, these two domain-docked conformations are thermodynamically different. First, domain docking dramatically favors only ATP binding, while ADP affinity is significantly less affected (*Figure 5B*). Moreover, only ATP but not ADP binding to the AMPylated V461F variant is entropically driven (*Figure 5—source data 2*), suggesting that domain docking is not responsible for the favorable entropic contribution to the free energy of ATP binding. This observation agrees well with the fact that ATP binding to the isolated BiP NBD* is also predominantly entropically driven (*Table 1*); but it is quite opposite to what has been previously observed for DnaK(*Zhuravleva et al., 2012*; *Zhuravleva and Gierasch, 2015*), where ATP-binding largely affects conformational dynamic of the SBD and thus, was entropically driven only for the two-domain protein, but not for the isolated NBD. We speculate that this difference in BiP and DnaK thermodynamics of ATP binding is implied by differences in communication between the NBD subdomains. It has been shown previously that the number of structural elements (i.e., salt bridges and a hydrophobic patch) connecting the NBD subdomains, varies between different Hsp70s and control the Hsp70 rate of nucleotide release (and their chaperone activity) (*Brehmer et al., 2001*). In DnaK, very strong communication between NBD subdomains apparently 'lock' the NBD in one ATP-bound conformation and thus, restrict its conformational flexibility. In turn, weaker communication between the BiP NBD subdomains results in the roughness of the NBD conformational landscape and enhanced NBD conformational flexibility in the presence of ATP.

## Discussion

The molecular mechanisms that underlie functional differences between different Hsp70s without significant structural variations between different family members have remained largely unknown. Whilst recent crystallographic descriptions of the ATP-bound states gave an impression that different Hsp70s adopt the same domain-docked and domain-undocked states, growing experimental evidence suggests that in solution each chaperone has a complex conformational landscape and co-exists as a heterogeneous ensemble of several conformations (*Zhuravleva et al., 2012*; *Lai et al., 2017*; *Marcinowski et al., 2011a*; *Mapa et al., 2010*; *Sikor et al., 2013*). In this study, we adopted the methyl NMR 'divide-and-conquer' approach to characterize thermodynamic and structural features of the conformational landscape for the ER Hsp70 chaperone BiP. Our characterization elucidated evolutionary features of this Hsp70 that help it to adjust to the ER environment and suggested the mechanism of post-translational regulations of BiP activity.

We obtained the in-depth characterization of the BiP conformational cycle and revealed several unique features of the BiP conformational landscape (*Figure 6*). As discussed above, the DnaK cycle comprises three functionally and structurally distinct conformations: (*Zhuravleva et al., 2012*; *Lai et al., 2017*) In the ADP-bound state, the chaperone adopts the linker-unbound, domain-undocked conformation (U); ADP-to-ATP exchange results in domain docking and predominant occupancy of the domain-docked conformation (D); and substrate-binding to the ATP-bound state results in domain undocking with population of an ensemble of the ADP-like (but ATP-bound) linker-unbound, domain-undocked conformation (U) and an allosterically active, intermediate linker-bound domain-undocked conformation (I) that has been shown to be essential for efficient substrate binding and release as well as ATPase hydrolysis (*Figure 6*, left).

The 'divide-and-conquer' analysis of methyl NMR spectra of the ATPase deficient variant of FL BiP and its NBD demonstrated that, despite the fact that BiP populates conformations structurally identical to DnaK, the distribution of the domain-docked and -undocked conformations for different ligand-bound states is drastically different (*Figure 6*, middle). Similar to DnaK, ATP binding to BiP favors NBD interactions with the interdomain linker and stabilizes domain docking. However, in contrast to DnaK, ATP-bound BiP co-exists in solution as an ensemble of two functional states: the domain-docked conformation (D) and an ADP-like (but ATP-bound) domain-undocked conformation (U), interconverting on the relatively slow (hundred millisecond–sub-second) time-scale. Substrate binding stabilizes the linker-unbound, domain-undocked conformation (U), while a small fraction of the domain-docked conformation (D) also co-exists in solution. Interestingly, interactions between the interdomain linker and NBD are only observed for the isolated NBD or for the domain-docked FL BiP, suggesting that the linker-bound conformation (I) is only transiently populated upon domain undocking. As expected, in the ADP-bound state, BiP adopts the linker-unbound, domain-undocked

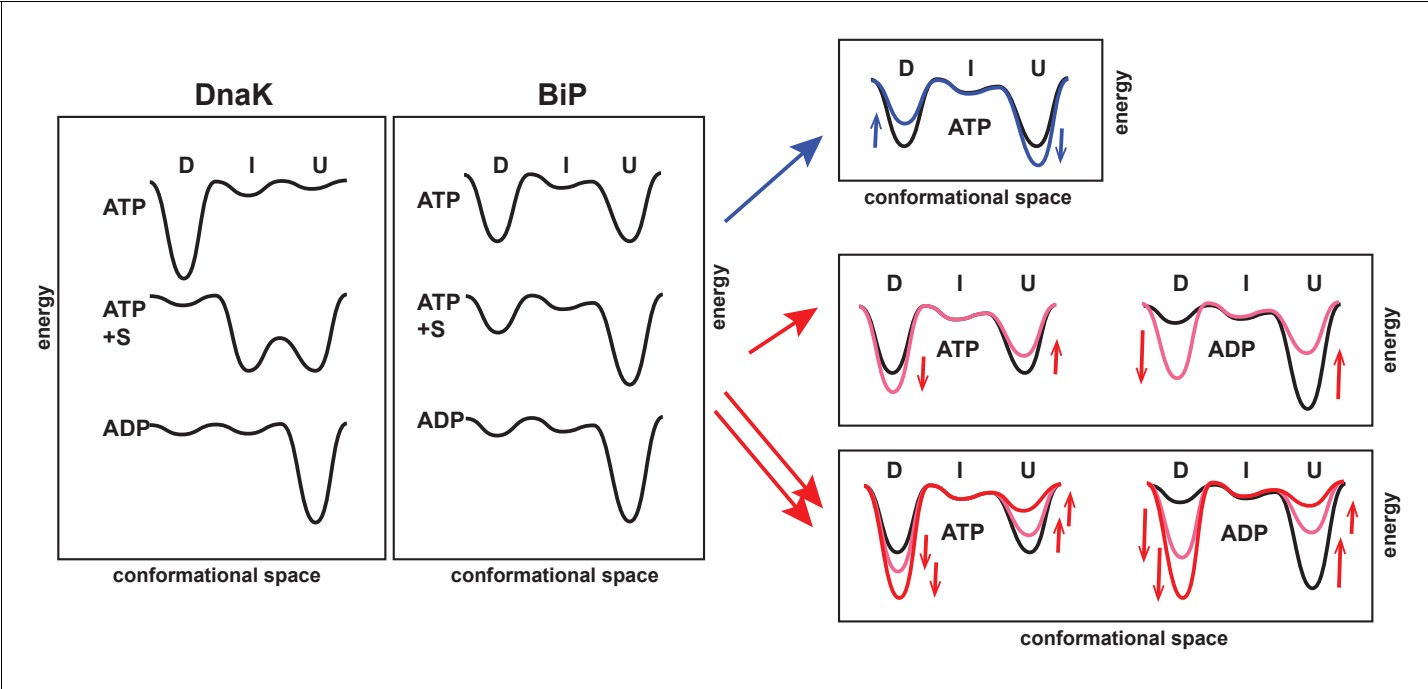

**Figure 6.** The unique features of the BiP conformational landscape and its posttranslational regulation: Schematic illustration of three key ligand-bound states (ATP-, ATP- and substrate-, and ADP-bound) of DnaK and BiP. For both chaperones the conformational landscapes comprises three structurally and functionally distinct conformations: domain-docked (D), intermediate domain-undocked linker-bound (I), and domain-undocked linker-unbound (U). Blue and red arrows illustrate how local perturbations of the SBD affect the BiP conformational landscape: the I437V and I538V substitutions favor domain undocking (blue arrows, *Figure 4*), while the V461F and I526V substitutions (*Figure 4*) as well as AMPylation of Thr518 (*Figure 5*) stabilize the domain-docked conformation in the presence of ATP and ADP. The additive effect, for example, upon the V461F substitution and AMPylation, is also shown (double red arrows).

DOI: https://doi.org/10.7554/eLife.29430.025

conformation (U). However, ATP binding is not essential for BiP to adopt the domain-docked conformation. This conformation is significantly populated upon AMPylation or other local perturbations of its βSBD even in the ADP-bound state, indicating that for BiP the linker-bound conformation is transiently populated even in the absence of ATP.

We hypothesize that the enhanced conformational heterogeneity of BiP enables post-translational regulation of BiP activity and function. Our findings illustrate that local perturbations of the BiP SBD via 'soft' single-site amino acid substitutions enable allosteric controlling of the BiP conformational ensemble (*Figure 6*, right). Moreover, the key regulation of BiP activity in vivo by AMPylation of Thr518 relies on the same allosteric mechanism: AMPylation of the SBD domain shifts the conformational equilibrium toward the domain-docked conformation either in the presence of ATP or ADP. This AMPylation-induced stabilization of the domain docking entirely rationalizes the experimentally observed effects of AMPylation on BiP function (Preissler S., Ron D., et al., personal communication, 2017, a co-submitted manuscript). As expected from AMPylation-induced stabilization of the docked state, which is inefficient for ATP hydrolysis and substrate binding, AMPylation promotes fast substrate release and prevents stimulation of ATP-hydrolysis by J-domain containing co-chaperones that observed experimentally by Ron and colleagues. Moreover, our allosteric model also explains enhanced protection of the interdomain linker from proteolysis and linker-promoted oligomerization observed for AMPylated BiP in the presence of both ADP and ATP (Preissler S., Ron D., et al., personal communication, 2017, a co-submitted manuscript).

Our results demonstrate that local perturbations in the BiP SBD fine-tune the pre-existing conformation ensemble populated upon binding to the certain ligands: For example, in the presence of ATP the I437V and I538V substitutions shift the thermodynamic equilibrium in the ensemble of the equally populated domain-docked and -undocked conformation toward the domain-undocked conformation (*Figure 6*, blue arrows), while the V461F and I526V substitutions (*Figure 4*) as well as

AMPylation of Thr518 favors the domain-docked conformation (*Figure 6*, red arrows). An evident correlation between the free energy of nucleotide binding and domain docking (*Figure 5B*) validates that populations of the domain-docked and -undocked conformations observed in NMR spectra indeed quantitatively represent thermodynamically relevant conformational distributions in the BiP functional ensemble.

Intriguingly, local perturbations in the BiP SBD may also overpower conformational effects of nucleotide binding without decoupling of inter-domain and intra-domain allosteric transitions. For example, the V461F substitution and AMPylation of Thr518 (*Figure 6*, red arrows) results in significant population of the domain-docked conformation even in the absence of ATP, whilst ATP binding has been previously suggested to be an essential step of domain docking in DnaK (*Zhuravleva et al., 2012*; *Zhuravleva and Gierasch, 2011*). Astonishingly, local SBD perturbations in the BiP SBD preserve the inter-domain allostery in BiP and may also complement each other (*Figure 6*). This possibility to combine effects of different SBD perturbations apparently provides an elegant tactic for the gradual real-time fine-tuning of BiP function *in vivo* by covalent post-translational modifications and interactions with co-chaperones and other members of the ER protein quality control system.

While mounting evidence suggests that BiP activity is linked to neurodegenerative diseases, diabetes, cardiovascular diseases, cancer progression and anticancer drug resistance, (*Schönthal, 2013*; *Dudek et al., 2009*) these tunable properties of the BiP conformational landscape are particularly interesting because they potentially provide new opportunities to develop BiP-specific allosteric drugs (*Nussinov and Tsai, 2013*) to enable precise modulations of BiP activity. Our findings also suggest that different members of the Hsp70 family apparently fine-tune their function evolutionally and post-translationally through adjustments of their conformational landscape rather than by altering chaperone structure. More broadly, our findings present experimental evidence that highlights the role of conformational dynamics in allosteric regulation (*Wrabl et al., 2011*; *Guo and Zhou, 2016*) of complex multidomain systems such as molecular chaperones.

## Materials and methods

### Expression and purification of FL BIP and its NBD

The hamster BiP with a non-cleavable N-terminal 6x-His-tag was cloned into the pET28a vector. The N-terminal His-tagged BiP construct has been extensively characterized in the literature and the His-tag does not perturb BiP function (*Wei and Hendershot, 1995*). We designed three constructs, each containing the T229G mutation, including full-length (FL) BiP(1-641), NBD(1-413), comprising NBD only and NBD(1-417), comprising NBD and four conserved residues from the interdomain linker. The 'soft' mutants I437V, V461F, I526V, and I538V were constructed to perturb key allosteric sites. The I145V and I371V mutants were also constructed to assign two peak doublets that were used to calculate populations of the domain-docked and –undocked conformations (see below).

The labeled and unlabeled protein was expressed in *E. coli* BL21(DE3) competent cells (Novagen). A single colony was resuspended in 2 ml LB broth (Fisher Scientific) supplemented with kanamycin and grown overnight at 37°C. The small amount of the overnight culture were transferred into 500 ml LB broth media to reach starting optical density at a wavelength of 600 nm ($OD_{600}$) ~0.1. The culture was incubated at 37°C with shacking until $OD_{600}$ reached ~0.9. Then protein expression was induced with IPTG (final concentration 1 mM) and the culture was grown for another 6 hr. Cells were harvested by centrifugation, resuspended in the binding buffer (20 mM HEPES, 400 mM NaCl, pH 8.0) and frozen at 80°C until purification step.

Expression of $^2$H, $^{15}$N, Ileδ1-[$^{13}$CH$_3$]-labeled samples were performed according to published method (*Tugarinov et al., 2006*) with minor modification. Particularly, transformants were grown in 2 ml LB broth for 6 hr, then cells were harvested and transferred to 25 ml M9 minimum media (starting $OD_{600}$ ~0.1) containing deuterated D-glucose (CIL, 1,2,3,4,5,6,6-D7, 98% DLM-2062), $^{15}$N-labeled ammonium chloride ($^{15}$NH$_4$Cl), 10% of deuterated Celtone Complete Medium (CIL, CGM-1040-D) and D$_2$O, and grew overnight at 30°C. The next day the pre-culture was transferred to 500 ml labeled M9 media (starting $OD_{600}$ ~0.2) and incubated at 37°C. When the $OD_{600}$ was reached ~0.7, 5 ml of methyl-$^{13}$C-labeled alpha-ketobutyric acid (CLM-6820) solution (14 mg/ml in

$D_2O$, pH 10.0) was added. After ~1 hr of incubation at 37°C, IPTG was added to the final concentration of 1 mM. The protein was expressed for 7 hr before harvesting as described above.

Thawed cells were incubated for 30 min on ice with 0.5 ml lysozyme (50 mg/ml, Sigma-Aldrich, UK), 0.1 ml DNaseI (10 mg/ml, Sigma-Aldrich) and Protease Inhibitor Cocktail (Roche). After sonication, lysates were spun down at 20000 rpm for 45 min and filtered to remove cell debris. Supernatants were loaded onto a HisTrap HP nickel column (GE Healthcare) pre-equilibrated with the binding buffer (20 mM HEPES, 400 mM NaCl, pH 8.0). Following a wash step with the binding buffer containing 40 mM imidazole, bound protein was eluted by 500 mM imidazole. Collected fractions were extensively dialysed against the final buffer (20 mM HEPES, 100 mM KCl, 5 mM $MgCl_2$, pH 7.6), flash frozen in liquid nitrogen and stored at −80°C. Protein concentrations were measured using UV absorbance at 280 nm. The purity of the samples were characterized using the SDS-PAGE gel and by measuring the A260/A280 ratio. The theoretical molar extinction coefficients of (28880 $M^{-1}$ $cm^{-1}$ for FL BIP, 17420 $M^{-1}$ $cm^{-1}$ for isolated NBD) were used for all calculations. Additionally, protein concentrations were measured using the Bradford protein assay (Bio-Rad) with bovine serum albumin as a standard.

## Expression and purification of the BiP SBD

The BiP SBD was obtained from the $^2H$, $^{15}N$, Ileδ1-[ $^{13}CH_3$]-labeled FL BiP using the BiP-linker specific protease SubA. 400 µl of 80 µM BiP was incubated for 4 hr at 30°C with 100 µl of 0.1 mg/ml SubA in the presence of 5 mM ATP. SBD was purified using one-step Ni-NTA affinity purification, dialyzed against the final buffer (20 mM HEPES, 100 mM KCl, 5 mM $MgCl_2$, pH 7.6) and concentrated to 20 µM.

SubA was expressed and purified using a standard protocol from Prof David Ron's laboratory (University of Cambridge). The protein was expressed in *E. coli* Origami 2 (DE3) cell (Novagen). A single colony was resuspended in 2 ml LB broth (Fisher Scientific) supplemented with ampicillin and grown overnight at 37°C. The small amount of the overnight culture were transferred into 500 ml LB broth media to reach starting optical density at a wavelength of 600 nm ($OD_{600}$) ~0.1. The culture was incubated at 37°C with shacking until $OD_{600}$ reached ~0.9. Them protein expression was induced with IPTG (final concentration 1 mM) and the culture was grown for another 3 hr. Cells were harvested by centrifugation, resuspended in the binding buffer (50 mM HEPES, 300 mM NaCl, 2 M Urea, 5% glycerol, pH 7.4) and frozen at 80°C until purification step.

Thawed cells were incubated for 30 min on ice with 0.5 ml lysozyme (50 mg/ml, Sigma-Aldrich), 0.1 ml DNaseI (10 mg/ml, Sigma-Aldrich) and Protease Inhibitor Cocktail (Roche). After sonication, lysates were spun down at 20000 rpm for 45 min and filtered to remove cell debris. Supernatants were loaded onto a HisTrap HP nickel column (GE Healthcare) pre-equilibrated with the binding buffer (50 mM HEPES, 300 mM NaCl, 2 M urea, 5% glycerol, pH 7.4). A gradual refolding of the protein was performed on-column using a linear urea gradient from 2M to 0M. Then, the protein was eluted using a linear imidazole gradient from 10 to 500 mM. Finally, the protein sample was dialysed against 50 mM HEPES, 300 mM NaCl, and 5% glycerol (pH 7.4).

## NMR experiments

To obtain fingerprints of the conformational states of the BiP functional cycle, isolated BiP NBD* constructs (residues 1–413 and residues 1–417) and FL BiP* constructs were expressed in isotopically labeled media as described above. NMR acquisitions were carried out in the NMR buffer (20 mM HEPES, 100 mM KCl, 5 or 40 mM $MgCl_2$, pH 7.6) in the presence 5 or 40 mM of ADP or ATP (no significant differences were observed in NMR spectra for different nucleotide and $MgCl_2$ concentrations) A BEST version (*Lescop et al., 2007*; *Schulte-Herbrüggen and Sorensen, 2000*) of amide transverse relaxation optimized spectroscopy (TROSY) (*Pervushin et al., 1997*) pulse sequence was used to record amide NMR spectra and a band-selective optimized-flip-angle short-transient experiment (*Schanda and Brutscher, 2005*) (2D $^1H$-$^{13}C$ SOFAST-HMQC) was used for methyl NMR (*Tugarinov et al., 2003*). All measurements were recorded at 25°C at 750, 900 and 950 MHz Bruker spectrometer equipped with a Bruker TCI triple-resonance cryogenically cooled probes. Data was processed with NMRPipe (*Delaglio et al., 1995*) and analyzed with CcpNmr Analysis software packages (*Vranken et al., 2005*; *Skinner et al., 2016*).

## Analysis of populations of the domain-docked and domain-undocked conformations

To assign peaks corresponded to domain-docked and domain-undocked conformations, the spectra of FL BiP constructs were overlaid with the spectra of the corresponding nucleotide-binding state of the isolated NBD*(1-413) and SBD. If we observed almost no changes in methyl NMR peak positions between the FL construct and the isolated NBD* and SBD (e.g., for ADP-bound BiP*, *Figure 2*, *Figure 4—figure supplement 1*), we assumed that there is no interdomain communication in the FL protein and all peaks in the FL spectrum were assigned to the domain-undocked (U) conformation. If two sets of peaks were observed in the NMR spectrum of a FL construct (e.g., for ATP-bound BiP, *Figure 2* and *Figure 4—figure supplement 1*); for each peak doublet, the peak that overlapped s perfectly with the corresponding peak from the spectrum of the isolated NBD was assigned to the domain-undocked (U) conformation. The second peak from the FL doublet that had distinct chemical shifts from the corresponding isolated NBD peak was assigned to the domain-docked (D) conformation.

To calculate the populations of the domain-docked and -undocked conformations from spectra of FL BiP*, we used methyl peak intensities of three non-overlapping peak doublets (shown in *Figure 2*, grey). Individual peaks in each doublet correspond to the domain-docked and -undocked conformations. The conformation for each peak was assigned by overlapping the spectra of FL BiP* with the spectra of isolated NBD*(1-413) as described above. We further assigned two peak doublets using the I145V and I371V single-point mutants. Both residues are located away from the nucleotide-binding site: Ile 145 is located near the NBD-αLid interface and directly reports on domain docking, while Ile 371 is located near the interface between NBD subdomains IIA and IIB, and thus reports on long-range conformational transitions in the NBD. Peak intensities were obtained using the parabolic method implemented in CcpNmr (*Vranken et al., 2005*; *Skinner et al., 2016*). The population of the domain docking conformation was calculated as: $p_D = I_D/(I_D +I_U)$x100%. Errors were calculated as standard deviations (SDs) from the means for three peak doublets.

## Isothermal Titration Calorimetry

All ITC experiments were performed on BiP* and its variants using MicroCal ITC200 system. The protein solution was dialyzed overnight against the ITC buffer (20 mM HEPES, 100 mM KCl, 5 mM $MgCl_2$, pH 7.6). The final protein concentrations were between 20–80 μM and the stock nucleotide concentrations varied between 0.1–0.4 mM. All measurements were carried out at 25°C. For each protein construct measurements were repeated at least three times; ΔH and $K_d$ values were obtained for each repeat using the non-linear least square curve-fitting algorithm implemented in MicroCal Origin (OriginLab, Northampton, MA). The ΔH and $K_d$ values were averaged out over all repeats and the corresponding TΔS and ΔG values were calculated using the following equitations:

$$\Delta G = RTLnK_d$$

$$-T\Delta S = \Delta G - \Delta H,$$

where R is the ideal gas constant and T is the temperature (298 K).

## Analysis of the thermodynamic linkage between domain docking and nucleotide binding

For each construct, the free energy of nucleotide (ATP or ADP) binding was obtained from the ITC data using the following equation: $\Delta G(binding)/RT = Ln(K_d)$, where $K_d$ is an ADP or ATP binding constant, R is the ideal gas constant, and T is the temperature in K. Populations of the domain-undocked and -docked conformations obtained from NMR spectra, were used to calculate the free energy of domain docking using the following equation: $\Delta G(docking)/RT = -Ln(p_D/(1 - p_D))$, where $p_D$ and 1-$p_D$ are populations of the domain-docked and -undocked conformations, respectively (*Figure 5—source data 2*).

To obtain theoretical plot of free energy of nucleotide (ADP or ATP) binding, $\Delta G(binding)/RT$, against the free energy of docking, $\Delta G(docking)/RT$, we assumed that the protein co-exists in solution as an ensemble of two conformations, domain-docked and -undocked, and both conformations

can bind to the corresponding nucleotide with different affinities. Thus, the free energy of nucleotide binding can be calculated using the following equation:

$$\Delta G(binding)/RT = Ln\left(p_D K_d^D + p_U K_d^U\right)$$

where $K_d^U$ and $K_d^D$ are the ADP or ATP binding constants of the 'pure' 100%-populated domain-docked and -undocked conformations, respectively; and $p_D$ and $p_U = 1\ p_D$ are the populations of the domain-docked and -undocked conformations, respectively. The free energy of docking were calculated:

$$\Delta G(docking)/RT = -Ln(p_D/(1-p_D))$$

where $p_D$ and $p_U = 1–p_D$ are the populations of the domain-docked and –undocked conformations, respectively.

## AMPylation

The AMPylation protocol was adopted from Preissler et al (*Preissler et al., 2015*).500 µL of 100 µM purified BiP* or BiP*-V461F were subjected to buffer exchange (25 mM HEPES-KOH pH 7.4, 100 mM KCl, 10 mM MgCl$_2$, 1 mM CaCl$_2$, 0.1% (v/v) Triton X-100) using NAP-5 columns (GE Healthcare) and AMPyltaed for 8 hr at 30°C with bacterially expressed GST-FICD-E234G at ratio 1:100 in the presence of 3 mM ATP. Next, AMPylated protein was subjected to the standard purification protocol described above. AMPylation, was validated by mass spectrometry.

## Liquid chromatography-mass spectrometry

The mass analysis was performed by LC-MS using an M-class ACQUITY UPLC (Waters UK, Manchester, UK) interfaced to a Synapt G2S Q-IMT-TOF mass spectrometer (Waters UK, Manchester, UK). 1 µL of 5 µM sample was loaded onto a MassPREP protein desalting column (Waters UK, Manchester, UK) washed with 10% solvent B (0.1% formic acid in acetonitrile) in solvent A (0.1% formic acid in water) for 5 min at 25 µL min$^{-1}$. After valve switching, the bound protein was eluted by a wash of 95% solvent B in A for 3 min before re-equilibration at 10% solvent B in A ready for the next injection. The column eluent was directed in to the mass spectrometer via a Z-spray electrospray source. The MS was operated in positive TOF mode using a capillary voltage of 3kV, sample cone of 40V and source offset of 80V. Backing pressure was 7.9 mbar and trap bias 2V. The source temperature was 80°C and desolvation was 150°C. Argon was used as the buffer gas at a pressure of $9.1 \times 10^{-3}$ mbar in the trap and transfer regions of the TriWave device. Mass calibration was performed by a separate injection of [Glu]-fibrinopeptide b at a concentration of 250 fmol µl$^{-1}$ in MS/MS mode and a CID voltage (trap region) of 28V. Data processing was performed using the MassLynx v4.1 suite of software (Waters UK, Manchester, UK) supplied with the mass spectrometer.

## Acknowledgements

This work was supported by BBSRC grant BB/M021874/1. The authors are grateful to Prof David Ron and Dr Steffen Preissler (University of Cambridge) for exchange of unpublished data and critical reading of the manuscript. Purified FICD and the SubA plasmid were a generous gift of Prof David Ron. We thank Prof Linda Hendershot (St. Jude Children's Research Hospital) for the WT BiP cDNA clone and Dr Jared Cartwright (Protein Production facility of the University of York) for help with cloning BiP to the pET28a vector. We thank Dr Arnout Kalverda (University of Leeds, bioNMR Facility), Dr Geoff Kelly (MRC Biomedical NMR Centre, The Francis Crick Institute, London, U.K.), and Dr Sara Whittaker (HWB-NMR, University of Birmingham) for assistance with NMR data collection; Dr Iain Manfield (Centre for Biomolecular Interactions, University of Leeds) for assistance with ITC measurements; Dr James Ault and Dr Rachel George (University of Leeds Mass Spectrometry Facility) for mass spectrometry data collection and analysis.

## Additional information

### Funding

| Funder | Grant reference number | Author |
|---|---|---|
| Biotechnology and Biological Sciences Research Council | BB/M021874/1 | Anastasia Zhuravleva |

The funders had no role in study design, data collection and interpretation, or the decision to submit the work for publication.

### Author contributions

Lukasz Wieteska, Data curation, Formal analysis, Investigation, Methodology, Writing—original draft; Saeid Shahidi, Data curation, Formal analysis, Validation, Methodology, Writing—original draft; Anastasia Zhuravleva, Conceptualization, Supervision, Funding acquisition, Validation, Investigation, Methodology, Writing—original draft, Project administration, Writing—review and editing

### Author ORCIDs

Anastasia Zhuravleva (iD) http://orcid.org/0000-0001-6165-8509

### Decision letter and Author response

Decision letter https://doi.org/10.7554/eLife.29430.027
Author response https://doi.org/10.7554/eLife.29430.028

## Additional files

### Supplementary files

• Transparent reporting form
DOI: https://doi.org/10.7554/eLife.29430.026

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
