## [Decision Letter]

Thank you for submitting your article "Allosteric fine-tuning of the conformational equilibrium poises the chaperone BiP for post-translational regulation" for consideration by *eLife*. Your article has been reviewed by two peer reviewers, and the evaluation has been overseen by a Reviewing Editor and Randy Schekman as the Senior Editor. The following individual involved in review of your submission has agreed to reveal his identity: Erik RP Zuiderweg (Reviewer #1).

The reviewers have discussed the reviews with one another and the Reviewing Editor has drafted this decision to help you prepare a revised submission.

Summary:

The conformational cycle of BiP, the Hsp70 chaperone of the endoplasmic reticulum (ER), is less well understood than that of the bacterial Hsp70, DnaK. Hsp70s consist of an N-terminal nucleotide binding domain and a C-terminal substrate binding domain and their activity relies on ATP- and protein substrate-controllable docking and undocking of these domains. Based on a methyl NMR analysis of BiP, this study reveals a surprising conformational heterogeneity of ATP-bound BiP that distinguishes BiP from DnaK. This property allows regulation of the BiP chaperone cycle by post-translational modifications such as AMPylation. Interestingly, BiP inactivation by AMPylation of its substrate binding domain stabilizes the domain-docked conformation independent of the specific nucleotide bound.

Major point to be addressed in the revised manuscript:

1) The following important conclusion is not represented in the manuscript, although it is supported by the data: The authors show, for the first time, that the NMR-observed conformations (docked vs. not-docked) are thermodynamically relevant in a quantitative way. This is validating earlier work from the Gierasch and other NMR labs, as well as establishing the base for future work by the authors. This conclusion should be stressed in the revised manuscript. This point refers to the following results:

"Next, to examine how AMPylation affects thermodynamics of nucleotide binding, we performed ITC measurements for the AMPylated and non-AMPylated V461F BiP* variant. ATP affinity increased from Kd of 800 nM for BiP* (that populated 53% of the domain-docked conformation) to Kd of 210 nM for the V461F variant (66% of the domain-docked conformation), and finally to Kd of 70 nM for the AMPylated V461F BiP (89% of the domain docked conformation) (Figure 5—figure supplement 5)."

Plotting the free energy of ATP binding for these 3 results versus the free energy of the Docking/Undocking percentages results in a very good correlation with a slope of 1.1. This suggests that the NMR-observed states are thermodynamically relevant in a quantitative way. It seems important to make this correlation clear in the Discussion, with the caveat that these chemical changes are of course not equilibrium thermodynamics. Thus, please include a Table and plot.

Other questions raised by the reviewers:

2) Too many figures of chemical shift differences are shown. Showing it once is sufficient.

3) Re. Materials and methods: Is the His tag on the C or N terminus? Was it cleaved off?

4) Re. Discussion: With respect to Table 1. The authors suggest that the large entropic contribution to the binding free energy of ATP to DnaK and to BiP is due to the release of the lid, and to increase of the SBD disorder. Fair enough for DnaK, where the entropy gain is much less for binding of ATP to the NBD alone. But not for BiP, where the binding of ATP to the NBD is still dominated by entropy gain. Any suggestions?

5) Why is the conversion between the D and U states in the ATP-bound conformation slow? Any idea?

6) Do the authors understand why the population of the U state is so much higher in BiP compared to DnaK?

7) Why is there a perfect resonance overlap in the isolated NBD and FL-BiP spectra in the D state? Wouldn't someone expect that in the docked state some NBD residues would be affected giving rise to different chemical shifts?

8) How were the D and U states assigned?

---

## [Author Response]

Major point to be addressed in the revised manuscript:1) The following important conclusion is not represented in the manuscript, although it is supported by the data: The authors show, for the first time, that the NMR-observed conformations (docked vs. not-docked) are thermodynamically relevant in a quantitative way. This is validating earlier work from the Gierasch and other NMR labs, as well as establishing the base for future work by the authors. This conclusion should be stressed in the revised manuscript. This point refers to the following results:"Next, to examine how AMPylation affects thermodynamics of nucleotide binding, we performed ITC measurements for the AMPylated and non-AMPylated V461F BiP* variant. ATP affinity increased from Kd of 800 nM for BiP* (that populated 53% of the domain-docked conformation) to Kd of 210 nM for the V461F variant (66% of the domain-docked conformation), and finally to Kd of 70 nM for the AMPylated V461F BiP (89% of the domain docked conformation) (Figure 5—figure supplement 5)."Plotting the free energy of ATP binding for these 3 results versus the free energy of the Docking/Undocking percentages results in a very good correlation with a slope of 1.1. This suggests that the NMR-observed states are thermodynamically relevant in a quantitative way. It seems important to make this correlation clear in the Discussion, with the caveat that these chemical changes are of course not equilibrium thermodynamics. Thus, please include a Table and plot.

We thank the reviewers for this constructive suggestion. As the reviewers suggested, we added the plot and table that show the correlation between the free energy of ATP binding and the free energy of domain docking for BiP* and its non-AMPylated and AMPylated V461F variant (Figure 5 and Figure 5—source data 2) in the revised version of the manuscript). Additionally, we made the same analysis for the free energy of ADP binding. The observed correlation between the free energies of nucleotide binding and domain docking was fitted using a two-state equilibrium model, in which the BiP co-exists as an ensemble of two thermodynamically distinct states, the domain-docked and -undocked. AMPylation and the V461F mutation shift the conformational equilibrium between these two states and thus, tune thermodynamics of nucleotide binding. We added a new Results section: “Post-translational fine-tuning of the BiP conformational equilibrium provides the precise thermodynamic control of nucleotide binding**”** as well as a new Materials and methods section: “Analysis of the thermodynamic linkage between domain docking and nucleotide binding”. As reviewers suggested, we also highlighted the fact that the observed correlation between thermodynamics of nucleotide binding and domain docking validates that BiP conformations observed by NMR indeed represent thermodynamically relevant conformational distributions in the BiP functional ensemble (Discussion, fifth paragraph).

Other questions raised by the reviewers:2) Too many figures of chemical shift differences are shown. Showing it once is sufficient.

As suggested, we reduced the amount of figures with NMR methyl spectra. Particularly, we removed Figure 4 and Figure 5 and C. We also modified Figure 2 (see below) to improve its clarity and conciseness.

3) Re. Materials and methods: Is the His tag on the C or N terminus? Was it cleaved off?

We thank the reviewers for pointing this out. In this study all FL BiP and NBD constructs have a noncleavable 6x-His-tag as it has been shown previously that the N-terminal His-tag does not perturb BiP function (Wei & Hendershot., JBC, 270(1995): 26670-26676. We added these details on Materials and methods (subsection “Expression and purification of FL BIP and its NBD”, first paragraph).

4) Re. Discussion: With respect to Table 1. The authors suggest that the large entropic contribution to the binding free energy of ATP to DnaK and to BiP is due to the release of the lid, and to increase of the SBD disorder. Fair enough for DnaK, where the entropy gain is much less for binding of ATP to the NBD alone. But not for BiP, where the binding of ATP to the NBD is still dominated by entropy gain. Any suggestions?

The reviewers raised a very interesting and important question! We don’t think that the current study provides enough experimental information to unambiguously answer this intriguing question. However, we speculate that weaker communication between NBD subdomains may cause the conformational heterogeneity of the ATP-bound NBD (please see a new Results section “Post-translational fine-tuning of the BiP conformational equilibrium provides the precise thermodynamic control of nucleotide binding”, last paragraph).

5) Why is the conversion between the D and U states in the ATP-bound conformation slow? Any idea?

To address this question, we estimated the time-scale of domain docking in BiP using slow exchange limit with the respect to the chemical shift differences between the two conformations. We found that the transition should be significantly slower than 50 ms, which is not completely unexpected, given that domain docking is a multistep process that requires very large structural rearrangements. We discussed this in the revised version of the manuscript: subsection “Conformational features of the BiP allosteric cycle”, third paragraph.

6) Do the authors understand why the population of the U state is so much higher in BiP compared to DnaK?

It has been previously demonstrated that even subtle (one amino-acid) perturbations in the DnaK sequence can cause very significant redistributions in the chaperone conformational ensemble (Zhuravleva, A., Clerico, E. M. and Gierasch, L. M. Cell 151, 1296-1307(2012)). To this end, one should expect very different populations of the U and D conformations for BiP, which share only ~60% sequence identity with DnaK. Consequently, the differences between DnaK and BiP are likely to rely on a cumulative effect from aminoacid differences between these two Hsp70s. Interestingly, our data revealed that the BiP conformational landscape has not only unique thermodynamic but also kinetic features. Particularly, the transition between the docked and undocked conformations is significantly slower in BiP as compared with DnaK. We described these observations in the revised manuscript, Subsection “Conformational features of the BiP allosteric cycle”, third and last paragraphs.

7) Why is there a perfect resonance overlap in the isolated NBD and FL-BiP spectra in the D state? Wouldn't someone expect that in the docked state some NBD residues would be affected giving rise to different chemical shifts?8) How were the D and U states assigned?

We apologize for the confusion. In the presence of ATP, two sets of peaks were observed in the NMR spectrum. For each peak doublet, one peak that overlaps perfectly with the corresponding peak from the spectrum of the isolated NBD, was assigned to the domain-undocked (U) conformation. The almost perfect overlapping between peaks for FL BiP and its isolated NBD suggests that the chemical environment in the FL protein and its isolated domain is very similar. On the contrary, the second peak from the FL doublet had distinct chemical shifts from the corresponding isolated NBD peak. This suggests that the second conformation of the ATP-bound FL BiP is significantly different from the conformation of the isolated NBD. We assigned these peaks to the domain-docked (D) conformation. We modified the Materials and methods (“Analysis of populations of the domain-docked and domain-undocked conformations” in the revised manuscript) and the legend for Figure 2. To further clarify and validate the assignments of the domaindocked and –undocked conformations, we produced the isolated SBD and recorded methyl NMR spectrum for this construct. We made a new version of Figure 2that shows now methyl NMR spectra of two main functional conformations of FL BiP, domain-docked and -undocked, overlaid with the spectra of the isolated NBD and SBD. As expected, the spectra of the isolated NBD and SBD were overlapped almost perfectly with the spectrum of ADP-bound, but not ATP-bound FL BiP. Finally, two of three doublets, which were used to calculate the populations of the domain-docked and -undocked conformations, were assigned to Ile residues 145 and 371 by mutagenesis.